# Organic Dye-Doped PMMA Lasing

**DOI:** 10.3390/polym13203566

**Published:** 2021-10-15

**Authors:** Pen Yiao Ang, Marko Čehovski, Frederike Lompa, Christian Hänisch, Dinara Samigullina, Sebastian Reineke, Wolfgang Kowalsky, Hans-Hermann Johannes

**Affiliations:** 1Institut für Hochfreuquenztechnik, Technische Universität Braunschweig, Schleinitzstraße 22, 38106 Braunschweig, Germany; pen.yiao.ang@ihf.tu-bs.de (P.Y.A.); marko.cehovski@ihf.tu-bs.de (M.Č.); f.lompa@tu-braunschweig.de (F.L.); wolfgang.kowalsky@ihf.tu-bs.de (W.K.); 2Dresden Intergrated Center for Applied Physics and Photonic Materials, Technische Universität Dresden, Nöthnitzer Straße 61, 01187 Dresden, Germany; christian.haenisch@tu-dresden.de (C.H.); dinara.samigullina@tu-dresden.de (D.S.); sebastian.reineke@tu-dresden.de (S.R.); 3Cluster of Excellence PhoenixD (Photonics, Optics, and Engineering—Innovation across Disciplines), 30167 Hannover, Germany

**Keywords:** organic laser, polymer laser, laser tuning

## Abstract

Organic thin-film lasers gain interest as potential light sources for application in diverse fields. With the current development, they hold variety of benefits such as: low-cost, high-performance, and color-tunability. Meanwhile, the production is not complicated because both the resonator and the gain medium can be assembled by solution-processable organic materials. To our knowledge, information about using poly(methyl methacrylate) (PMMA) as a matrix for organic dye lasers was insubstantial. Herein, the feasibility of using organic dye-doped PMMA as an organic dye laser was tested. Six different sample designs were introduced to find out the best sample model. The most optimum result was displayed by the sample design, in which the gain medium was sandwiched between the substrate and the photoresist layer with grating structure. The impact of dye concentration and grating period on peak wavelength was also investigated, which resulted in a shift of 6 nm and 25 nm, respectively. Moreover, there were in total six various organic dyes that could function well with PMMA to collectively perform as ‘organic dye lasers’, and they emitted in the range of 572 nm to 609 nm. Besides, one of the samples was used as a sensor platform. For instance, it was used to detect the concentration of sugar solutions.

## 1. Introduction

Organic dye laser is a challenging research topic because it provides a new horizon for simple, low-cost, time-saving, versatile, and environmental-friendly fabrication of new and desirable laser structures. Diverse application fields have been found, such as lab-on-chip spectroscopy [1], data/optical communication [2], refractive index sensor [3], vapour pressure detector [4], absorption and transmission spectroscopy [5,6]. Distributed-feedback (DFB) lasers are fabricated by introducing grating structure to organic dye-doped lasers, and this type of lasers can provide single-mode emission with narrow bandwidth. Generally, two different methods are followed to design a DFB polymer thin-film laser. Firstly, a grating structure can directly be written into the polymer/gain medium using electron-beam lithography or holographic lithography [7,8,9,10,11,12]. Secondly, the gain material can either be solution-processed (e.g., spin-coating and dip-coating) or vacuum deposited on the grating structure [13,14,15,16,17,18]. The lasing wavelength of a DFB thin-film laser can be easily tuned by applying electrical control [19], modulating the thickness of the active layer [20,21], stretching the active film [22,23,24], designing a wedge-shaped film [12,25], applying different grating periods [26,27] and using a liquid crystal [28,29].

To function as an organic gain media, organic dyes have been dissolved in liquid solvents for the lasing purpose for decades [30]. However, the organic liquid laser can induce unnecessary evaporation because of their solvent. To solve this problem, organic solid-state lasers can be generated. Generally, the gain medium is categorised into two main groups. First group includes the polymer which is a fluorophore and it can directly be used as a gain medium [31,32,33]. Second group of gain medium requires doping of several organic dyes into non-conjugated polymers. By using this method, the devices can be fabricated to be mechanically flexible and the preparation can be non-expensive by using solution-based methods. The typically applied polymers that are mentioned in the literature are polystyrene (PS) and PMMA. Most of these studies proposed different DFB thin-film architectures by using PS as a matrix [34,35]. On the other hand, PMMA was mostly found to be optical fiber [36,37,38] or fiber laser [39,40]. Thin-film based on dye-doped PMMA [41,42] showed a Full-Width-at-Half-Maximum (FWHM) as big as 10 nm. Literature regarding the integration of grating structure on organic dye-doped PMMA film can be barely found.

In this study, we demonstrate the feasibility of organic dye-doped PMMA thin-film laser by analysing different sample designs with PMMA as a matrix by using the solution-processed method, spin-coating. In the results section, constriction of FWHM of the spectra and clear lasing threshold are shown. Modifications of the gain medium and the resonator were carried out to demonstrate the lasing state. Furthermore, the workability of a set of 6 different alternative organic dyes was evaluated. Moreover, the application potential as a refractive index sensor was presented as well.

## 2. Results

### 2.1. Design Variation

To find out the sample model with the best lasing performance, six different sample designs were introduced. Rhodamine 6G (Rh6G) was selected as the first dye dopant in this experiment because it is commonly found in the literature for PMMA fiber laser [43,44,45]. The concentration of Rh6G in PMMA was fixed at 400 ppm. In the first design (D1), a flat and single-layer of dye-doped PMMA was spin-coated on the Si-Substrate. Next, a grating structure was proposed directly on the layer of dye-doped PMMA in the second design (D2). In the third design (D3), a layer of photoresist, EpoClad was generated on top of the dye-doped PMMA thin-film. Furthermore, an EpoClad layer was also spin-coated on top of the dye-doped PMMA with a grating structure in the fourth design (D4). In the fifth design (D5), a flat dye-doped PMMA layer was sandwiched between the substrate and the grating structured EpoClad layer. In the sixth design (D6), a layer of EpoClad with a grating structure was produced on top of the substrate and covered by a spin-coated dye-doped PMMA layer. D6 is commonly shown in various publications [7,13,15,21,32].

When the grating structure was introduced to the sample design, the required grating periods were calculated by using the following equation:(1)$$\lambda_{Bragg}=\left(\frac{2n_{eff}\Lambda }{m}\right)$$
where $n_{eff}$ is the effective refractive index of dye-doped PMMA, *m* is the order of diffraction and Λ is the grating period. $\lambda_{Bragg}$ is the resonant wavelength in the cavity, which is reinforced during its propagation along the active layer before it is diffracted in the grating across different directions. Second order of diffraction (*m* = 2) was applied here.

Other than this, the grating period of both PMMA layer and EpoClad layer was intended to be 380 nm as per the peak wavelength of the emission spectrum of Rh6G. Meanwhile, the measured average grating period on top of the PMMA layer was 379.85 nm, while it was 380.16 nm on top of the EpoClad layer (see Figure 1). The result was significantly close to the desired value. Therefore, it can be deduced that the grating structure with the expected grating period can be properly built on top of the PMMA layer and the EpoClad layer.

The result of lasing properties due to design variation is shown in Figure 2. The photoluminescence spectrum of the Rh6G doped sample has a FWHM of about 36 nm. By using D1, the FWHM was reduced to 9.01 nm. According to the study of Samuel, I.D. et al. [46], the sample achieved the state of amplified stimulated emission (ASE) because values of FWHM of ASE are typically around 10 nm. Therefore, it could be said that D1 reached ASE but not lasing state. With the application of the grating structure, the FWHM of D2 was only 1.16 nm. Meanwhile, by using D3, the FWHM was 2.90 nm. This meant that the grating structure had a strong effect on the constriction of FWHM compared to the EpoClad layer. However, when the EpoClad layer was generated on top of the grating structure, the FWHM of D4 became 2.18 nm. This indicates that EpoClad could not improve the FWHM when the grating structure was directly applied on the gain medium. Among the samples, D5 had the smallest FWHM (0.83 nm). This design was suggested by Zhai, T. et al. [31], who proposed the idea that direct structuring on the gain medium would cause uneven distribution of light modes along the gain medium. With D5, the light mode was distributed completely and it helped in improving the lasing properties. By using D6, the FWHM became slightly larger (1.97 nm), which could be caused by the direct interaction of the gain medium and the excitation source. In such a way, the light could not be confined well in the gain medium.

For the aspect of peak wavelength, D1 and D3 were found to have nearly same value. This is because the designs of D1 and D3 were nearly the same and both had the same layer thickness of Rh6G doped PMMA. On the other hand, D2 and D4 had the same peak wavelength because the layer thickness of dye-doped PMMA of both designs was the same. D5 and D6 had a thicker dye-doped PMMA layer, therefore the peak wavelength was detected at a longer wavelength. Hence, it can be deduced that the peak wavelength was red-shifted to a longer wavelength with an increasing layer thickness of dye-doped PMMA. There are more dye dopants with thicker layer thickness. There is a high possibility of reabsorption of the emission of the dopants. The red-shift might be caused by the re-emission of the reabsorbed photons.

The lasing threshold of the samples was investigated by exciting them with a series of pump energies. The turning point, in which the light was amplified, was noticed as the lasing threshold. By using D1, the lasing threshold was the lowest among the designs 0.131 μJ. This was due to the fact that there was no additional optical system like a grating structure or an EpoClad layer. With the application of a grating structure or an EpoClad layer like in D2 or D3, the lasing threshold increased on a small scale (approx. 0.2– 0.3 μJ). When the combination of the grating and EpoClad layer was applied like in D4, D5, and D6, the lasing threshold raised further (> 0.3 μJ). It can be deduced here that the lasing threshold increased when the grating and/or EpoClad layer was implemented.

By comparing the polarisation extinction ratio (PER) between the samples, the value of D1 was the smallest (21.30 dB). The result of D2 (24.57 dB) and D3 (24.53 dB) showed that the grating structure and the EpoClad layer helped in improving the PER with the same strength. With the combination of the grating structure and the EpoClad layer, the PER can be further improved like in D4 (28.12 dB) and D5 (30.88 dB). D5 had the most optimum PER because the distribution of light mode along the gain medium was thorough without disturbance of the grating structure. By using D6, the PER (23.49 dB) did not show the highest value because the gain medium was in direct contact with the excitation source.

Among the samples, D2 had the lowest output energy (1.38 nJ), which was due to the fact that the layer thickness of PMMA was the narrowest. With a flat thin-film, D1 had an output energy of 3.63 nJ, which is higher than that of D2’s because the PMMA layer of D1 was thicker. Therefore, the thicker the gain medium, the higher the output energy. By adding an EpoClad layer on it like in D4, the output energy was improved to 6.00 nJ. With an additional EpoClad layer on the pure thin-film like in D3, the output energy was enhanced to 8.56 nJ. Here, it can be deduced that an additional EpoClad layer can help in improving the output energy, by providing extra reflectivity due to the difference of refractive index. D5 had the highest output energy (16.43 nJ) because the light was amplified well through the grating structure and the EpoClad layer. On the other hand, by using D6, the output energy was 8.82 nJ because there was no extra optical structure on top of the gain medium for the light amplification.

Even though the lasing threshold of D5 is not the smallest, it was chosen as the best sample model because it provided the best lasing performance. D5 had the smallest FWHM (0.83 nm), the highest PER (30.88 dB), and the highest output energy (16.43 nJ). The peak wavelength of D5 was detected at 575 nm. For these aforementioned reasons, D5 was selected for further experiments.

### 2.2. Concentration Variation

Any changes in the gain medium will affect the lasing properties. To assure the laser state, the property of the gain medium was manipulated. Here, three different concentrations of Rh6G were examined, which were 100 ppm, 200 ppm, and 400 ppm, while the grating period was set at 380 nm. The result is displayed in Figure 3. The FWHM of 100 ppm Rh6G doped sample was 10.85 nm, which indicated that only ASE were attained and no laser state was reached because 100 ppm concentration was too low, thus fewer dye molecules could contribute to laser generation. When the concentration of Rh6G was increased to 200 ppm and 400 ppm, the FWHM was <1 nm, which were 0.89 nm and 0.83 nm, respectively. At the same time, the peak wavelengths of 200 ppm and 400 ppm were 569 nm and 575 nm, subsequently. There was a 6 nm red-shift of peak wavelength when the Rh6G concentration was increased from 200 ppm to 400 ppm. It could be explained that by the presence of more dyes in a higher concentration, which had a higher reabsorption possibility and induced the red-shift eventually.

By using a 100 ppm Rh6G doped sample, no lasing threshold can be seen. This confirms again that the sample did not achieve the lasing state. The lasing threshold of the 200 ppm Rh6G doped sample ( 0.530 μJ) was higher than the 400 ppm Rh6G doped sample. This is due to the fact that there was less dye in 200 ppm compared to 400 ppm. Therefore, more energy was needed to obtain lasing state. The same explanation can be used to understand the trend of the output energy. The higher the concentration is, the higher would be the output energy. When the concentration was higher, there were more dye molecules in the sample, which could contribute to the output energy. For the aspect of PER, the value was about >28 dB, once the sample attains the lasing state.

### 2.3. Grating Variation

On top of the changes of the gain medium, any changes of the resonator will alter the lasing properties. Another method to confirm the lasing state was to modify the resonator. In this case, the grating structure was the resonator of D5. Three different grating periods (370 nm, 380 nm, and 390 nm) were applied to check its influence on the lasing properties, while the Rh6G concentration was fixed at 400 ppm. The result is illustrated in Figure 4. A 25 nm wavelength-shift has been observed by using the three aforementioned grating periods. The measured peak wavelengths were 564 nm, 575 nm, and 589 nm, respectively. The FWHM of all the spectra was found to be <1 nm. On top of this, it has also been observed that the lasing threshold decreased when a smaller grating period was applied. The reason could be that the optical gain increased with a smaller grating period (see Figure 4a). Meanwhile, the higher the optical gain, the higher the stimulated emission cross section [47]. Miniscalco, W. et al. suggested that the lasing threshold became smaller when the stimulated emission cross section was higher [48]. On the other hand, the output energy became higher with increasing stimulated emission cross section, when the applied grating period was smaller. Besides, it is observed that the PER was quite high (>28 dB), which was not strongly affected by the changes of the grating period.

### 2.4. Dyes Variation

The result shown so far was obtained by using the Rh6G doped sample. It is important to check the workability of other organic dyes together with PMMA. It has been observed that six different organic dyes showed positive results, which were Rh6G, Rhodamine B (RhB), Lumogen Orange (LumO), Pyrromethene 597 (P597), 4-(Dicyanomethylene)-2-tert-butyl-6-(1,1,7,7-tetramethyljulolidin-4-yl-vinyl)-4H-pyran (DCJTB) and 4-(Dicyanomethylene)-2-methyl-6-julolidyl-9-enyl-4H-pyran (DCM2). The concentration of these dyes was set at 400 ppm, while the applied grating period was calculated according to Equation (Equation 1) using the peak wavelength of the respective photoluminescence spectra (see Appendix A). The result is elaborated in Figure 5. The lasing spectra of these dye-doped samples were found to be in the range from 572 nm to 609 nm. All the FWHMs were found to be <1 nm. Meanwhile, all the samples had relatively high PER (>28 dB). Among the samples, P597 has a peak wavelength of 572 nm with the smallest FWHM (0.57 nm), the lowest lasing threshold ( 0.253 μJ), the highest output energy (18.04 nJ), and the longest sample lifetimes (33,000 pump pulses).

### 2.5. Application Potential

Since the P597 doped sample showed the best performance, the sample model was modified by attaching a cover to it for application possibility (see Figure 6). In this way, solutions with different sugar concentrations were injected into it for checking their influence on the lasing spectra. With increasing sugar concentration, the peak wavelength was red-shifted to a longer wavelength. When the sugar concentration was more than 20 wt%, smaller peaks with lower intensity were detected. This could be due to the injected sugar which tended to re-crystallise because of the increasing sugar concentration. A high sugar concentration corresponds to a high refractive index. It was observed that, when the sugar concentration increased, the refractive index went up (see Appendix A), and the peak wavelength of the lasing spectra increased exponentially. This indicated that the sample can be used as a refractive index sensor caused the sugar concentration.

## 3. Conclusions

In this work, the feasibility of using PMMA as a matrix with a grating structure to function as a thin-film laser was proven. At first, six different sample designs were introduced to find out the best sample model as dye-doped PMMA thin-film laser. The design, in which the gain medium was sandwiched between the substrate and the EpoClad layer with grating structure, provided the best performance. Moreover, the lasing state of the sample was re-established by alternating the gain medium and the resonator, in which the dye concentration and the grating period were varied. By using the Rh6G concentrations from 100 ppm to 400 ppm, a peak wavelength shift of 6 nm is observed. Meanwhile, by modifying the grating periods from 370 nm to 390 nm, a peak wavelength-shift of 25 nm was also observed. Besides, a total six various dyes were confirmed to function well together with PMMA as organic dye lasers, namely, Rh6G, RhB, LumO, P57, DCJTB, and DCM2. The peak wavelengths of these different dye-doped samples were in the range of 572 nm to 609 nm. With the best dye option, the P597 dye-doped PMMA sample was modified to examine solutions with different sugar concentrations, by observing the shift of peak wavelength according to the changes of the refractive index of the sugar concentrations. Therefore, it can be concluded that the workability of dye-doped PMMA as a thin-film laser with application potential is confirmed.

## 4. Methods and Materials

PMMA used in this research, PLEXIGLAS^®^ 8N, was purchased from the company, Röhm GmbH, Darmstadt, Germany. It was supplied in uniform size as pellets. To dope different dyes in PMMA, PMMA pellets together with the dyes were dissolved in butanone (purchased from Sigma-Aldrich Chemie GmbH, Taufkirchen, Germany). The concentration of the PMMA concerning to butanone was kept at 7.5 wt%, while the concentration of the dyes was determined with respect to PMMA. Rh6G, RhB, P597, and DCJTB were purchased from Radiant, Wermelskirchen, Germany, while LumO and DCM2 were purchased from TCI, Tokyo, Japan. EpoClad (purchased from micro resist technology GmbH, Berlin, Germany) was diluted with gamma-Butyrolactone into 50 wt% before being used.

Since both PMMA and EpoClad were solution-processable materials, the technique of spin-coating was applied. To generate a grating structure directly on the PMMA surface, a stamp with the appropriate grating structure was placed on top of the PMMA thin-film. Together with the stamp, the sample was heated at 130 ${}^{{}^{\circ}}C$ under a pressure of 6.125 mN/mm${}^{2}$ for 25 min.

To cure EpoClad, an EpoClad layer was spin-coated in advance. Then, the EpoClad layer was pre-baked at 120 ${}^{{}^{\circ}}C$ for 5 min. Afterwards, it was cured under UV light for 20 s. Then, it was post-baked at 120 ${}^{{}^{\circ}}C$ for 3 min. The progress was preformed together with a stamp if the grating structure was established on top of the EpoClad layer.

To measure the optical gain, the variable stripe length method was applied. The setup is demonstrated in Figure 7a. A Nd:YAG laser (STA-01-1-1SH purchased from Stand, Vilnius, Lithuania) was implemented. The pump intensity was controlled by a Glan-Thomson polariser. Before reaching the sample, the pump intensity passed through a few lenses and aperture to assure a homogenous excitation line (see Figure 7b). The sample surface from an edge was excited perpendicularly to avoid the losses due to total internal reflection. For the optical gain measurement, the length of excitation was varied to avoid light saturation.

For the measurement of lasing properties, the setup was identical to the setup of the variable stripe length method. The length of excitation was fixed at 2.3 mm. The emission from the sample was collected for evaluation. The detector used was either a spectrometer (Triax320 purchased from Jobin Yvon, Bensheim, Germany) or a powermeter (LabMax™-TOP purchased from Coherent, Santa Clara, CA, USA). For the measurement of PER, the rotating polariser method was used. A polariser (LPVISE100-A purchased from Thorlabs, Bergkirchen, Germany) was established in front of the detector. Spectra were taken at 0° and 90°. The maximum values of each spectrum were compared using the following equation:(2)$$PER=10log_{10}\frac{P_{max}}{P_{min}}.$$

## Figures and Tables

**Figure 1 polymers-13-03566-f001:**
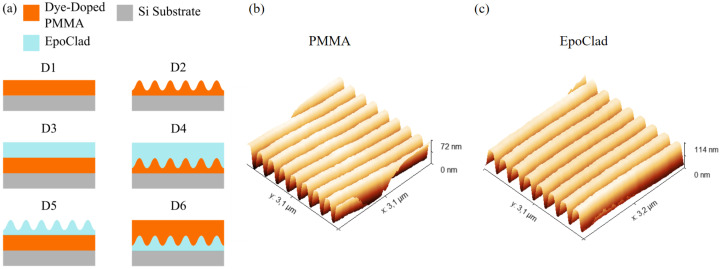
(**a**) Six different sample designs were produced to find out the best sample model. The grating profile on the (**b**) PMMA layer and the (**c**) EpoClad layer were observed.

**Figure 2 polymers-13-03566-f002:**
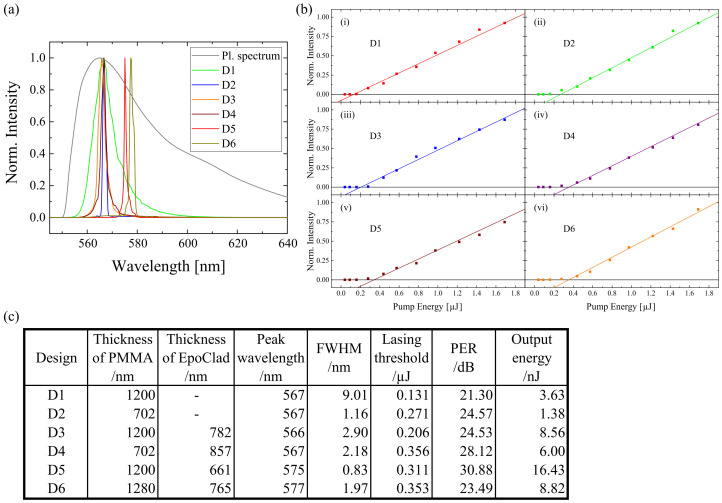
(**a**) The photoluminescence spectrum of Rh6G doped thin-film and the lasing spectra of design variation were shown. (**b**) The lasing threshold of different sample designs were evaluated. An overview of the lasing properties under the influence of the changes of sample design was presented in (**c**).

**Figure 3 polymers-13-03566-f003:**
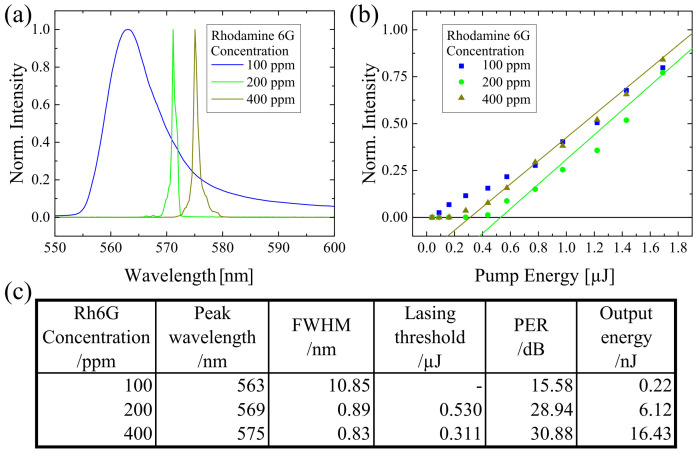
By using different Rhodamine 6G concentrations (100 ppm, 200 ppm, and 400 ppm), (**a**) the lasing spectra and (**b**) the lasing threshold were determined. (**c**) A table was presented to provide an overview of the lasing properties under the influence of Rhodamine 6G concentration variation.

**Figure 4 polymers-13-03566-f004:**
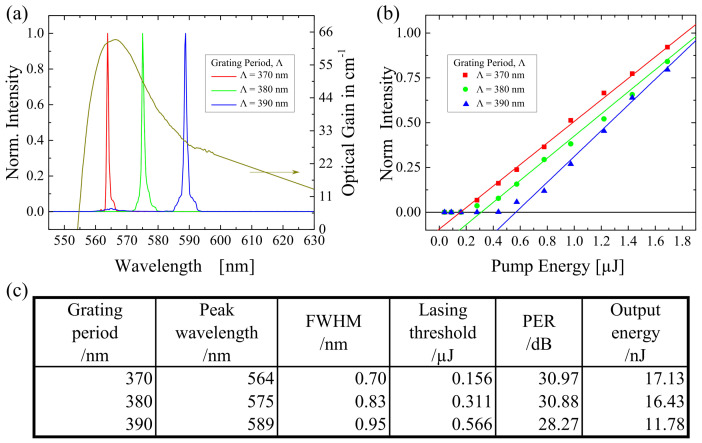
When the grating period (370 nm, 380 nm, and 390 nm) was altered, (**a**) the lasing spectra and (**b**) the lasing threshold were defined. While, the summary of the result of lasing properties was presented in (**c**).

**Figure 5 polymers-13-03566-f005:**
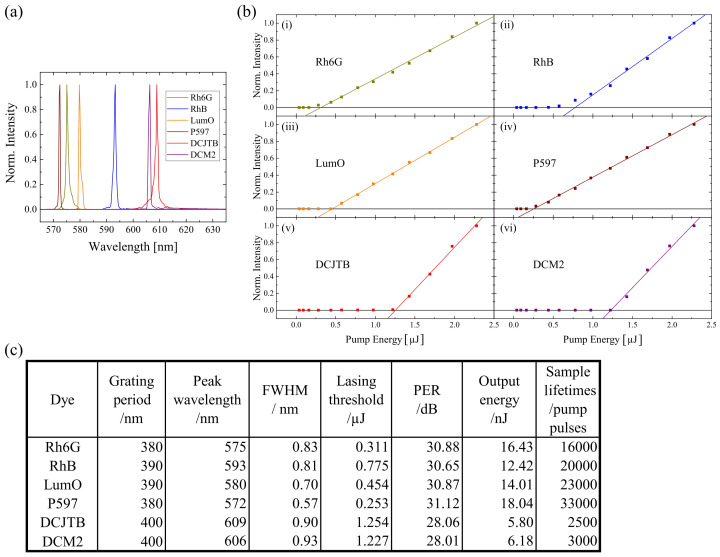
Six different organic dyes were found to be good dopants for PMMA. (**a**) The lasing spectra and (**b**) the lasing threshold of the respective dye-doped PMMA were measured. Meanwhile, an overview of the result was demonstrated in (**c**).

**Figure 6 polymers-13-03566-f006:**
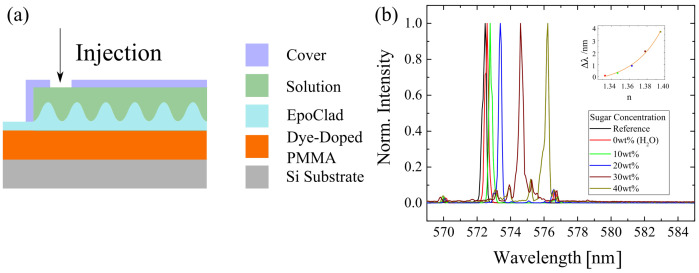
(**a**) The sample model was modified to detect the solution with different sugar concentration. (**b**) The lasing spectra under the influence of different sugar concentration were determined.

**Figure 7 polymers-13-03566-f007:**
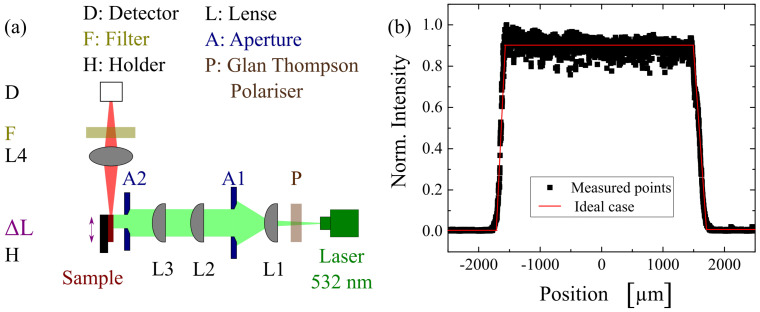
(**a**) The setup of the optical gain measurement and the measurement of lasing properties, and (**b**) the intensity profile of the pump light were shown.

## Data Availability

The data in this work are available upon request from the corresponding author.

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
