# Peer review of "Organic Dye-Doped PMMA Lasing"

_polymers, 2021, doi:10.3390/polym13203566_

Round 1

Reviewer 1 Report

  1.  plagiarism is 5 % see attached file 
  2. abbreviations should be defined at the first appeared time
  3. the conclusion needs more details
  4.  

Author Response

First, we would like to thank the reviewer for the comments on our manuscript. We were able to address the comments in the manuscript. The changes are as follows

Reviewer 1:

  • Spell check is accomplished.
  • The writing construction and the grammar is corrected.
  • Misspelt terms are corrected
  • The conclusion is modified to include all the information.

Reviewer 2 Report

The manuscript “Organic dye-doped PMMA lasing” summarized the study of organic dye-doped PMMA as an organic dye laser. They created a number of varieties of gratings used as resonators. The structures included PMMA doped with various dyes and were utilised as active media and it was shown that the Nd: YAG laser was pumped with 532 nm wavelength and part of the structure could emit lasing. The result appears to be correct and well presented and the methods described are adequate.

Generally speaking, this text is well written. I believe the readers will be attracted to it. After the following remarks, it can be published on this journal

At the first time appears, abbreviations should be specified

Some small errors occur in the text, a spell check is necessary.

Some terms are misspelt like "PMMA layer and EpoClad layer should be 380 nm depending on the maximum wavelength of the radiation range Rh6G."

More information are needed in the conclusion

Author Response

First, we would like to thank the reviewer for the comments on our manuscript. We were able to address the comments in the manuscript. The changes are as follows

Reviewer 2:

  • Spell check is accomplished.
  • Abbreviations are specified at the first appearance.
  • The writing construction and the grammar is corrected.
  • The conclusion is modified to include all the information.

Reviewer 3 Report

In the manuscript “Organic dye-doped PMMA lasing”, the authors reported a study of organic dye-doped PMMA as an organic dye laser.

They developed several types of grating structures, which were used as resonators. The structures contained PMMA doped with different dyes, used as active media, and it was demonstrated that some of the structures can emit lasing when were pumped with the 532 nm wavelength of an Nd: YAG laser. 

The authors proved the application potential of the device, by using it as a sensor platform to detect the concentration of sugar solutions. They obtained tunability of the peak wavelength by increasing the sugar concentration.

The result seems correct and well presented, and the methods are adequately described.

There are some minor mistakes in the text, spell check is required.

Some nouns seem to be missing a determiner before it. Consider adding an article.

Some words are misspelt, such as  “PMMA layer and EpoClad layer was intended to be 380 nm according the peak wavelength of the emission spectrom of Rh6G”.

Some constructions are wordy and they are repeating in the same phrase. Such as:  The concentration of the PMMA with respect to butanone was kept at 7.5wt%, while the concentration of the dyes was determined with respect to PMMA. Rh6G, RhB, P597, and…  (Methods and Materials section).

Thereby, I recommend this manuscript for publication in Polymers, Polymer Applications section, with minor revision.

Author Response

First, we would like to thank the reviewer for the comments on our manuscript. We were able to address the comments in the manuscript. The changes are as follows

  • Spell check is accomplished.
  • The writing construction and the grammar is corrected.
  • The conclusion is modified to include all the information

Reviewer 4 Report

Reviewer Report

In this manuscript, the authors report on organic dye-doped PMMA lasing. The manuscript reports interesting and important results in the field, so my recommendation is to accept the manuscript for publication, subject to the following revision points:

Since PMMA is commonly used for fabrication of optical fibers, which are used in optical communication and sensory systems, some relevant works in that area should also be cited, for example:

A. Simovic, A. Djordjevich, B. Drljaca, S. Savovic, R. Min, Investigation of bandwidth in multimode graded index plastic optical fibers, Optics Express, Vol. 29, No. 19, 2021, pp. 29587-29594.

S. Savovic, B. Drljaca, M. S. Kovacevic, A. Djordjevich, J. S. Bajic, D. Z. Stupar, G. Stepniak, Frequency response and bandwidth in low NA step index plastic optical fibers, Applied Optics, Vol. 53, No. 30, 2014, pp. 6999-7003.

A. Simovic, A. Djordjevich, B. Drljaca, S. Savovic, Wavelength dependent equilibrium mode distribution and steady state distribution in W-type plastic optical fibers with graded index core distribution, Optik, Vol 246, 2021, 167775 (9 pp).

P. Han, L. Li, H. Zhang, L. Guan, C. Marques, S. Savovic, B. Ortega, R. Min, X. Li, Low-cost plastic optical fiber sensor embedded in mattress for sleep performance monitoring, Optical Fiber Technology, Vol. 64, 2021, 102541 (8pp).

The mentioned works should be along with a further citation of polymer optical fiber random lasers, such as:

J. He, Wing-Kin E. Chan, X. Cheng, Ming-Leung V. Tse, C. Lu, Ping-Kong A. Wai, S. Savovic, Hwa-Yaw Tam, Experimental and theoretical investigation of coherent polymer optical fiber random laser, Advanced Optical Materials, Vol. 6, No. 7, 2018, pp. 1701187 (1-9).

Finally, authors should mention in the manuscript whether they tried to perform any modeling of the organic dye-doped PMMA lasing, such as Monte Carlo method-based numerical modeling which can describe the lasing dynamics (please refer to ref. He et al., AOM, Vol. 6, 2018).

Author Response

First, we would like to thank the reviewer for the comments on our manuscript. We were able to address the comments in the manuscript. The changes are as follows

  • Spell check is accomplished.
  • The writing construction and the grammar is corrected.
  • The conclusion is modified to include all the information.
  • The suitable suggestions of citation are included.